# Association of Bone Mineral Density and Coronary Artery Calcification in Patients with Osteopenia and Osteoporosis

**DOI:** 10.3390/diagnostics10090699

**Published:** 2020-09-16

**Authors:** Tzyy-Ling Chuang, Malcolm Koo, Yuh-Feng Wang

**Affiliations:** 1Department of Nuclear Medicine, Dalin Tzu Chi Hospital, Buddhist Tzu Chi Medical Foundation, Chiayi 62247, Taiwan; b8601139@tmu.edu.tw; 2School of Medicine, Tzu Chi University, Hualien City, Hualien 97004, Taiwan; 3Department of Nursing, Tzu Chi University of Science and Technology, Hualien City, Hualien 97005, Taiwan; m.koo@utoronto.ca; 4Center of Preventive Medicine, Dalin Tzu Chi Hospital, Buddhist Tzu Chi Medical Foundation, Chiayi 62247, Taiwan

**Keywords:** bone mineral density, coronary artery calcification, osteopenia, osteoporosis

## Abstract

The aim of this study was to investigate the association between bone mineral density (BMD) and coronary artery calcification (CAC) in adults with osteopenia or osteoporosis. A retrospective medical review study was conducted in a regional hospital in southern Taiwan. Medical records of patients who underwent both a coronary computed tomography scan and a BMD measurement were identified. Multinomial logistic regression analyses were used to assess the association between BMD and CAC levels in patients with osteopenia or osteoporosis. Of the 246 patients, 119 were female and 42.3% had CAC. For patients with osteopenia, after adjusting for the significant factors of CAC, no significant association was observed between BMD with either moderate CAC (0 < CAC score ≤ 100) or high CAC (CAC score > 100). However, in patients with osteoporosis, after adjusting for the significant factors of CAC, BMD in the lumbar spine was inversely associated with moderate CAC (odds ratio = 0.38, *p* = 0.035). In conclusion, a lower BMD in the lumbar spine was associated with an increased risk of moderate CAC in patients with osteoporosis. It is crucial to take action to maintain bone health, particularly in those who already have osteoporosis, to reduce the risk of developing CAC and its associated morbidity and mortality.

## 1. Introduction

Coronary artery calcification (CAC), a type of vascular calcification occurring in coronary arteries, plays an important role in the pathogenesis of coronary atherosclerosis [1]. It can act as a surrogate marker for total calcified plaque burden in both subclinical and overt atherosclerosis [2]. The pathophysiology is similar for vascular calcification and osteogenesis [3,4]. Vascular calcification has been linked to dysregulated bone and mineral metabolism, reduced bone mineral density (BMD), and increased fracture rates [5,6,7]. A meta-analysis of 25 studies including 10,299 patients concluded that low BMD was significantly associated with a higher incidence of atherosclerotic vascular abnormalities [8]. Another meta-analysis evaluating the association between BMD, vascular calcification, and the risk of osteoporosis and osteopenia revealed that patients with vascular calcification have significantly lower lumbar spine and hip BMD levels and increased risk of developing osteoporosis and osteopenia [9]. However, the association between BMD and CAC in patients with osteopenia and osteoporosis has not been reported. Therefore, the aim of the study was to explore the risk of CAC associated with BMD in patients with osteopenia or osteoporosis.

## 2. Materials and Methods

### 2.1. Study Design and Patients

The protocol of this study was approved on May 3, 2019 by the Institutional Review Board of the Dalin Tzu Chi Hospital, Buddhist Tzu Chi Medical Foundation, Taiwan (No. B10802011). We retrospectively identified eligible patients using the computerized medical record system of a regional teaching hospital in southern Taiwan. The inclusion criteria included adult patients who underwent both coronary computed tomography (CT) scan and BMD measurement during their general health examination from May 2014 to December 2019. The two assessments must have been performed within one month of each other. The exclusion criteria included patients whose BMD measured areas contained metal.

### 2.2. Coronary Artery Calcification Score

A CAC score was obtained from unenhanced axial images scanned before the coronary CT angiography for each patient. The scans were performed using a multidetector CT system (LightSpeed VCT, GE Medical Systems, GE Healthcare, Chicago, IL, USA). CAC was quantified with the Agatston scoring method [10]. The total calcium score was determined using the sum of individual scores from the four major coronary arteries (left main, left anterior descending, circumflex, and right coronary arteries).

### 2.3. Measurement of Bone Mineral Density

BMD was assessed by dual-energy X-ray absorptiometry (DXA) using a Discovery Wi DXA system (Hologic Inc., Marlborough, MA, USA). Absolute BMD values and T-scores (number of standard deviations (SDs) below the BMD of a young normal reference group, Asia database) were calculated for all patients. The measured areas included the lumbar spine and bilateral hips (total and femoral neck regions). The same densitometer was used for all patients to ensure consistent comparisons. According to the World Health Organization classification, osteopenia and osteoporosis were defined as a T-score of <−1.0 SD to > −2.5 SD and ≤ −2.5 SD, respectively. According to the International Society for Clinical Densitometry and International Osteoporosis Foundation for the diagnosis of osteoporosis and osteopenia, participants were designated as osteoporotic if they had a T-score ≤ 2.5 at either the hip or the spine.

### 2.4. Measurement of Demographic and Clinical Variables

Information on demographic variables was ascertained by face-to-face interview or questionnaire on the same day of the BMD assessment. The demographic variables included age, sex, body mass index, and smoking history while medical history included hypertension, diabetes mellitus, and hyperlipidemia. In addition, blood samples were also collected on same day to obtain data of high-density lipoproteins, fasting blood glucose, calcium, alkaline phosphatase, and estimated glomerular filtration rate (eGFR). Systolic blood pressure was also recorded.

### 2.5. Statistical Analysis

Results were expressed as mean and SD, or number and percentage, as appropriate. Differences in means and frequency distribution were compared using one-way analysis of variance or Chi-squared test, respectively. Multinomial logistic regression was used to evaluate the independent factors associated with CAC. CAC was categorized into three groups: no CAC (CAC score = 0), moderate CAC (0 < CAC score ≤ 100), high CAC (CAC score > 100). In addition, the risk of CAC associated with BMD, adjusting for potential confounding factors, was assessed separately for patients with osteopenia and osteoporosis. In the multivariable analyses, laboratory data (e.g., blood pressure) was evaluated instead of its associated disease (e.g., hypertension) to minimize the effects of disease treatment on disease status and physiological condition. Odds ratios (ORs) were expressed as the effect of 1 SD increase in the values of the respective independent variable. All statistical analyses were performed using PASW Statistics 18 (SPSS Inc., Chicago, IL, USA).

## 3. Results

### 3.1. Characteristics of All Patients

Of the 246 patients (127 males and 119 females), 104 (42.3%) had coronary artery calcification (CAC > 0). Participants with a high CAC were significantly older and more likely to have hypertension, diabetes mellitus, and hyperlipidemia, compared to those with moderate or no CAC. In addition, significantly higher systolic blood pressure and fasting blood glucose were observed in those with a high CAC. A significantly lower level of high-density lipoprotein was observed in those with moderate CAC compared with the no CAC group. A significantly lower eGFR was observed in the group with high CAC compared with the other two groups (Table 1). The distribution of the right and the left femoral neck T-score in patients with hypertension or diabetes mellitus is included in the Appendix A (Appendix A).

### 3.2. Patients with Osteopenia

Of the 194 patients with osteopenia, 83 (42.8%) had coronary artery calcification (CAC score > 0). Results from the multinomial logistic regression analysis of the independent variables are shown in Table 2. Age (OR = 1.68), body mass index (OR = 1.49), male sex (OR = 4.37), smoking (OR = 3.21), hypertension (OR = 2.77), and systolic blood pressure (OR = 1.59) were significantly associated with an increased risk of moderate CAC. Conversely, high-density lipoprotein (OR = 0.67) was significantly and inversely associated with an increased risk of moderate CAC. Moreover, age (OR = 4.67), male sex (OR = 4.11), hypertension (OR = 8.78), diabetes mellitus (OR = 10.14), hyperlipidemia (OR = 2.94), systolic blood pressure (OR = 1.54), and fasting blood glucose (OR = 1.74) were significantly associated with an increased risk of high CAC. Conversely, high-density lipoprotein (OR = 0.66) and eGFR (OR = 0.51) were significantly and inversely associated with an increased risk of high CAC.

Table 3 shows the results of the multinomial logistic regression analysis of CAC with BMD and T-score in patients with osteopenia, adjusting for age, body mass index, sex, smoking, systolic blood pressure, high-density lipoproteins, fasting blood glucose, and eGFR. No significant associations were observed for any of the BMD measures or T-scores with either moderate CAC or high CAC.

### 3.3. Patients with Osteoporosis

Of the 52 patients with osteoporosis, 21 (40.4%) had coronary artery calcification (CAC score > 0). Results from the multinomial logistic regression analysis of the independent variables are shown in Table 4. None of the independent variables were significantly associated with moderate CAC. Only age (OR = 4.64), hypertension (OR = 11.25), and systolic blood pressure (OR = 2.58) were significantly associated with an increased risk of high CAC.

Table 5 shows the results of the multinomial logistic regression analysis of the association of CAC with BMD and T-score in patients with osteoporosis, adjusting for age and systolic blood pressure. The BMD of the lumbar spine was significantly and inversely associated with moderate CAC (OR = 0.38, *p* = 0.035). The T-score of the lumbar spine was inversely associated with moderate CAC with marginal significance (OR = 0.44, *p* = 0.051). However, no significant associations were observed between high CAC and any BMD measures or T-scores.

## 4. Discussion

CAC plays an important role in the development of coronary artery disease and is a predictor of cardiovascular risk and mortality [11,12]. A CAC score of 0 in patients with suspected stable angina has a high negative predictive value for the exclusion of obstructive coronary artery disease and is associated with a good medium-term prognosis [13]. The CAC scores and major adverse cardiac events are strongly and positively correlated in patients with stable angina [14]. Two recognized types of CAC are intimal and medial artery calcification [15]. Atherosclerotic calcification mainly occurs in the intima [16]. CAC in the media is associated with advanced age, diabetes, and chronic kidney disease [17]. Several studies have confirmed that advanced age, diabetes, hyperlipidemia, hypertension, male sex, smoking, and renal disease are risk factors of intimal calcification, whereas renal dysfunction, hypercalcemia, hyperphosphatemia, parathyroid hormone abnormalities, and duration of dialysis are related to medial calcification [18].

The presence and extent of CAC can be predicted by traditional cardiovascular risk factors, including age, male sex, body mass index, ethnicity, hypertension, diabetes, hyperlipidemia, smoking, obesity, and family history of coronary artery disease [19,20]. Findings from this study showed that high CAC was also associated with age, male sex, hypertension, diabetes, and hyperlipidemia in patients with osteopenia. In addition, systolic blood pressure, high-density lipoproteins, fasting blood glucose, and eGFR were also found to be significantly associated with high CAC. Previous studies showed that high systolic blood pressure is associated with CAC, as well as with the risk of coronary and cardiovascular events [21,22]. Patients with subclinical atherosclerosis, as determined by CAC scores, reportedly have fewer high-density lipoprotein particles [23]; conversely, increasing levels of high-density lipoproteins were also associated with less CAC [24]. Impaired fasting glucose level was also an independent risk factor for CAC [25]. Stage III to V chronic kidney disease has been associated with increased CAC scores [26], and such patients had more frequent and more severe CAC [27]. In addition, a decrease in eGFR was found to be significantly associated with an increased CAC in individuals above 70 years of age [28]. Furthermore, although diabetes is an important marker for severe CAC, we did not include it in the multiple regression model. Instead, we evaluated fasting blood glucose, systolic blood pressure, and high-density lipoproteins for their potential confounding effects. The latter continuous variables were used because they were obtained from laboratory measurements, whereas diseases were collected based on self-report by the patients. We believe the use of laboratory results would be associated with a lower risk of misclassification.

A number of studies have shown an association between BMD and CAC, but others found little or no such association [29,30,31,32]. A study of 467 Korean women recruited from routine medical checkups noted that low BMD of the femur and the lumbar spine were associated with the coronary calcium score after adjusting for age [33]. In addition, the Copenhagen General Population Study with 2548 participants showed that BMD and CAC were inversely associated in both men and postmenopausal women [34]. Another study of 5590 patients at risk of coronary artery disease showed that lower BMD levels were significantly associated with CAC, particularly in postmenopausal women, and that low BMD levels combined and a CAC score > 0 was an independent risk factor for mortality [35]. Conversely, a cohort study of 490 middle-aged women reported that volumetric BMD was not associated with moderate or high CAC [36]. Another study of 366 middle-aged and older adults not using hormone therapy found no age-independent association between CAC and BMD at any site [37]. It is plausible that the association between coronary and bone calcium might be mediated by estrogen, but further studies are required to elucidate the underlying mechanism.

In the present study, only lumbar spine BMD and T-score were associated with moderate CAC in patients with osteoporosis. However, no association between CAC and BMD was observed in patients with osteopenia. This difference might be explained by the shift of noncalcified plaque to calcified plaque with increasing age, which could affect the vulnerability of these lesions over time [38]. In fact, the mean age of our patients with osteoporosis was 59.5 years (SD = 9.7 years) and was significantly older than the 56.4 years (SD = 9.0 years) in those with osteopenia (*p* = 0.032). This study also confirmed the complex regulatory networks of CAC. Multiple risk factors for CAC, including age, body mass index, sex, smoking, hypertension, diabetes mellitus, hyperlipidemia, systolic blood pressure, high-density lipoproteins, fasting blood glucose, and eGFR, were significantly associated with moderate CAC or high CAC. When these factors were applied to the multivariable analysis in patients with osteoporosis, only age and systolic blood pressure remained independently associated with CAC. For patients with osteopenia, after adjusting for other independent factors of CAC, including age, body mass index, sex, smoking, systolic blood pressure, high-density lipoproteins, fasting blood glucose, and eGFR, neither BMD nor T-score in all sites showed a significant association with either moderate CAC or high CAC.

The pathophysiology is similar between vascular calcification and osteogenesis [3,37]. Vascular calcification has been linked to dysregulated bone and mineral metabolism, reduced BMD and increased fracture rates [5,6,7]. Vascular calcification is recognized as an active process involving an interaction between inducers and inhibitors [4]. It also represents a complex biological process of calcium phosphate deposition and is related to the regulation of osteogene expression, bone morphogenetic protein, calcification inhibitors, and inflammatory cytokines [39,40].

There are some limitations in this study worth noting. First, this is a retrospective medical review study based on data from relatively healthy individuals undergoing health examination. Therefore, the number of patients with osteoporosis was limited. Second, we had no information on the use of medications in our participants. Nevertheless, laboratory data, instead of their associated diseases were used to minimize the treatment effects on disease status and physiological condition of the study participants. Third, BMD was measured by DXA rather than quantitative computed tomography (QCT). While DXA is cheaper and more easily performed, QCT has greater diagnostic sensitivity than DXA in detecting osteoporosis [41]. Fourth, all data were obtained from a single regional hospital in southern Taiwan, which may limit the generalizability of the findings.

## 5. Conclusions

Findings from this retrospective medical record review study indicated that the risk of moderate CAC was inversely and independently associated with BMD of the lumbar spine in adults with osteoporosis. Although CAC has complex regulatory networks, once osteoporosis develops, CAC needs to be taken care of. It is crucial to take action to maintain bone health, particularly in those who already have osteoporosis, to reduce the risk of developing CAC and its associated morbidity and mortality.

## Figures and Tables

**Table 1 diagnostics-10-00699-t001:** Demographic and clinical characteristics of the participants (*n* = 246).

Variable	No CAC (Score = 0)	Moderate CAC (Score = 1–100)	High CAC (Score ≥ 101)	*p*
*n* (%)	142 (57.7)	62 (25.2)	42 (17.1)	
Age, mean (SD), years	54.3 (8.6) ^a^	58.2 (8.4) ^b^	64.6 (7.9) ^c^	<0.001
Female (%)	84 (59.2)	22 (35.5)	13 (31.0)	<0.001
Smoking (%)	22 (15.5)	19 (30.6)	6 (14.3)	0.028
Hypertension (%)	18 (12.7)	15 (24.2)	24 (57.1)	<0.001
Diabetes mellitus (%)	6 (4.2)	3 (4.8)	12 (28.6)	<0.001
Hypertension and diabetes mellitus (%)	4 (2.8)	1 (1.6)	8 (19.0)	0.001
Hyperlipidemia (%)	11 (7.7)	6 (9.7)	8 (19.0)	0.108
Body mass index, mean (SD), kg/m^2^	24.1 (3.0)	25.1 (2.7)	24.8 (3.7)	0.066
Systolic blood pressure, mean (SD), mmHg	123.4 (21.8) ^a^	128.8 (20.6) ^ab^	134.1 (22.0) ^b^	0.012
High-density lipoprotein, mean (SD), mg/dL	52.0 (14.5) ^a^	46.6 (13.5) ^b^	46.8 (13.1) ^ab^	0.015
Glucose, mean (SD), mg/dL	102.7 (16.4) ^a^	107.1 (17.7) ^ab^	113.4 (25.4) ^b^	0.004
Calcium, mean (SD), mmol/L	2.2 (0.1)	2.2 (0.1)	2.2 (0.1)	0.506
Alkaline phosphatase, mean (SD), IU/L	73.4 (19.0)	72.5 (21.8)	72.0 (20.4)	0.903
eGFR, mean (SD), mL/min/1.73 m^2^	87.4 (16.2) ^a^	85.9 (22.0) ^a^	76.6 (16.1) ^b^	0.003
L-spine BMD, mean (SD), g/cm^2^	0.883 (0.114)	0.881 (0.139)	0.911 (0.137)	0.392
L-spine T-score, mean (SD)	−1.2 (1.0)	−1.2 (1.2)	−1.0 (1.1)	0.558
Right femoral neck BMD, mean (SD), g/cm^2^	0.642 (0.076)	0.656 (0.089)	0.660 (0.085)	0.296
Right femoral neck T-score	−1.6 (0.6)	−1.4 (0.9)	−1.5 (0.7)	0.239
Right hip total BMD, mean (SD), g/cm^2^	0.784 (0.104)	0.810 (0.116)	0.818 (0.104)	0.113
Right hip total T-score	−0.8 (0.7)	−0.7 (0.8)	−0.7 (0.8)	0.802
Left femoral neck BMD, mean (SD), g/cm^2^	0.652 (0.080)	0.674 (0.096)	0.661 (0.083)	0.242
Left femoral neck T-score	−1.5 (0.7)	−1.4 (0.8)	−1.5 (0.7)	0.288
Left hip total BMD, mean (SD), g/cm^2^	0.763 (0.100)	0.789 (0.100)	0.790 (0.109)	0.134
Left hip total T-score	−0.0 (0.8)	−0.9 (0.7)	−1.0 (0.7)	0.843

Note: Means with different superscripts indicate significant difference at *p* < 0.05 level, adjusted using Sidak post-hoc test. CAC: coronary artery calcification; SD: standard deviation; eGFR: estimated glomerular filtration rate; BMD: bone mineral density.

**Table 2 diagnostics-10-00699-t002:** Multinomial logistic regression analysis of the association of demographic and clinical variables with CAC in patients with osteopenia.

Variable	Odds Ratio (95% Confidence Interval) [*p*]		
	No CAC (Score = 0)	Moderate CAC (Score = 1–100)	High CAC (Score > 100)
*n* (%)	111 (57.2)	49 (25.3)	34 (17.5)
Age	1.00	1.68 (1.13–2.50) [0.011]	4.67 (2.63–8.32) [<0.001]
Body mass index	1.00	1.49 (1.05–2.11) [0.026]	1.32 (0.89–1.96) [0.166]
Sex (ref. = female)	1.00	4.37 (2.03–9.43) [<0.001]	4.11 (1.71–9.88) [0.002]
Smoking	1.00	3.21 (1.48–6.99) [0.003]	1.19 (0.43–3.29) [0.745]
Hypertension	1.00	2.77 (1.20–6.39) [0.017]	8.78 (3.65–21.13) [<0.001]
Diabetes mellitus	1.00	0.90 (0.17–4.82) [0.904]	10.14 (3.21–32.00) [<0.001]
Hyperlipidemia	1.00	1.58 (0.53–4.72) [0.411]	2.94 (1.00–8.61) [0.049]
Systolic blood pressure	1.00	1.59 (1.12–2.24) [0.009]	1.54 (1.04–2.27) [0.029]
High-density lipoproteins	1.00	0.63 (0.43–0.91) [0.014]	0.66 (0.43–1.00) [0.049]
Fasting blood glucose	1.00	1.41 (0.97–2.05) [0.075]	1.74 (1.18–2.56) [0.005]
Calcium	1.00	0.98 (0.70–1.38) [0.905]	0.79 (0.54–1.16) [0.226]
Alkaline phosphatase	1.00	0.85 (0.60–1.20) [0.363]	0.80 (0.53–1.19) [0.269]
eGFR	1.00	0.74 (0.52–1.06) [0.099]	0.51 (0.33–0.78) [0.002]

CAC: coronary artery calcification; eGFR: estimated glomerular filtration rate. Odds ratio was expressed as a one standard deviation increase in the independent variable (age = 9.01 years, body mass index = 2.90 kg/m^2^, systolic blood pressure = 20.22 mmHg, high-density lipoproteins = 14.29 mg/dL, fasting blood glucose = 18.68 mg/dL, eGFR = 16.55 mL/min/1.73 m^2^, calcium = 0.09 mg/dL, and alkaline phosphatase = 18.44 IU/L).

**Table 3 diagnostics-10-00699-t003:** Multinomial logistic regression analysis of the association of BMD and T-score with CAC in patients with osteopenia.

Variable	Adjusted Odds Ratio (95% Confidence Interval) [*p*]		
	No CAC *n* = 111 (Score = 0)	Moderate CAC *n* = 49 (Score = 1–100)	High CAC *n* = 34 (Score > 100)
Lumbar spine BMD	1.00	0.98 (0.65–1.46) [0.912]	1.17 (0.71–1.92) [0.543]
Lumber spine T-score	1.00	0.99 (0.67–1.47) [0.969]	1.12 (0.69–1.84) [0.644]
Right femoral neck BMD	1.00	1.20 (0.81–1.80) [0.367]	1.21 (0.73–2.02) [0.465]
Right femoral neck T-score	1.00	1.22 (0.83–1.81) [0.314]	1.20 (0.73–1.99) [0.470]
Right total hip BMD	1.00	1.06 (0.70–1.62) [0.785]	1.05 (0.63–1.75) [0.844]
Right total hip T-score	1.00	1.11 (0.73–1.67) [0.635]	1.09 (0.67–1.77) [0.732]
Left femoral neck BMD	1.00	1.40 (0.96–2.05) [0.085]	1.09 (0.67–1.78) [0.717]
Left femoral neck T-score	1.00	1.45 (0.99–2.10) [0.055]	1.10 (0.68–1.80) [0.691]
Left total hip BMD	1.00	1.03 (0.69–1.55) [0.870]	0.95 (0.57–1.61) [0.860]
Left total hip T-score	1.00	1.08 (0.72–1.62) [0.703]	0.96 (0.57–1.62) [0.884]

BMD: bone mineral density; CAC: coronary artery calcification; eGFR: estimated glomerular filtration rate. All models were adjusted for age, sex, body mass index, smoking, systolic blood pressure, high-density lipoproteins, fasting blood glucose, and eGFR. Odds ratio was expressed as a one standard deviation increase in the independent variable (lumbar spine BMD = 0.104 g/cm^2^, lumbar spine T-score = 0.9, right femoral neck BMD = 0.065 g/cm^2^, right femoral neck T-score = 0.5, right total hip BMD = 0.093 g/cm^2^, right total hip T-score = 0.6, left femoral neck BMD = 0.070 g/cm^2^, left femoral neck T-score = 0.6, left total hip BMD = 0.087 g/cm^2^, and left total hip T-score = 0.6).

**Table 4 diagnostics-10-00699-t004:** Multinomial logistic regression model of demographic and clinical variables for CAC in patients with osteoporosis.

Variable	Odds Ratio (95% Confidence Interval) [*p*]		
	No CAC (Score = 0)	Moderate CAC (Score = 1–100)	High CAC (Score > 100)
*n* (%)	31 (59.6)	13 (25.0)	8 (15.4)
Age	1.00	1.84 (0.82–4.12) [0.138]	4.64 (1.51–14.24) [0.007]
Body mass index	1.00	1.17 (0.61–2.25) [0.630]	1.15 (0.53–2.51) [0.718]
Sex (ref. = female)	1.00	0.44 (0.08–2.42) [0.348]	1.47 (0.29–7.47) [0.645]
Smoking	1.00	0.43 (0.05–4.13) [0.467]	NC
Hypertension	1.00	0.56 (0.06–5.58) [0.623]	11.25 (1.91–66.39) [0.008]
Diabetes mellitus	1.00	2.50 (0.14–43.29) [0.529]	4.29 (0.24–77.22) [0.324]
Hyperlipidemia	1.00	NC	2.07 (0.16–26.22) [0.574]
Systolic blood pressure	1.00	0.73 (0.36–1.45) [0.363]	2.58 (1.02–6.55) [0.046]
High-density lipoproteins	1.00	0.84 (0.43–1.65) [0.618]	0.82 (0.36–1.85) [0.628]
Fasting blood glucose	1.00	1.17 (0.56–2.43) [0.678]	1.49 (0.75–2.98) [0.254]
Calcium	1.00	1.30 (0.69–2.43) [0.418]	1.04 (0.44–2.46) [0.935]
Alkaline phosphatase	1.00	1.38 (0.70–2.72) [0.356]	1.52 (0.71–3.24) [0.277]
eGFR	1.00	1.55 (0.81–2.95) [0.187]	0.39 (0.12–1.29) [0.121]

CAC: coronary artery calcification; NC: not calculable; eGFR: estimated glomerular filtration rate. Odds ratio was expressed as a one standard deviation increase in the independent variable (age = 9.69 years, body mass index = 3.18 kg/m^2^, systolic blood pressure = 27.15 mmHg, high-density lipoproteins = 13.73 mg/dL, fasting blood glucose = 19.42 mg/dL, eGFR = 23.29 mL/min/1.73 m^2^, calcium = 0.12 mg/dL, and alkaline phosphatase = 23.75 IU/L).

**Table 5 diagnostics-10-00699-t005:** Multinomial logistic regression analysis of the association of BMD and T-score with CAC in patients with osteoporosis.

Variable	Odds Ratio (95% Confidence Interval) [*p*]		
	No CAC *n* = 31 (Score = 0)	Moderate CAC *n* = 13 (Score = 1–100)	High CAC *n* = 8 (Score > 100)
Lumbar spine BMD	1.00	0.38 (0.16–0.93) [0.035]	1.02 (0.40–2.60) [0.964]
Lumber spine T-score	1.00	0.44 (0.19–1.01) [0.051]	0.97 (0.38–2.45) [0.945]
Right femoral neck BMD	1.00	0.75 (0.36–1.55) [0.438]	1.24 (0.49–3.14) [0.655]
Right femoral neck T-score	1.00	1.22 (0.60–2.51) [0.586]	0.90 (0.33–2.45) [0.839]
Right total hip BMD	1.00	1.04 (0.51–2.13) [0.908]	2.37 (0.83–6.77) [0.106]
Right total hip T-score	1.00	0.98 (0.46–2.10) [0.961]	1.40 (0.58–3.38) [0.457]
Left femoral neck BMD	1.00	0.86 (0.42–1.77) [0.685]	1.17 (0.45–3.00) [0.750]
Left femoral neck T-score	1.00	0.96 (0.48–1.93) [0.911]	0.94 (0.37–2.38) [0.903]
Left total hip BMD	1.00	1.34 (0.67–2.69) [0.413]	2.22 (0.82–6.04) [0.117]
Left total hip T-score	1.00	1.27 (0.63–2.53) [0.505]	1.42 (0.57–3.53) [0.451]

BMD: bone mineral density; CAC: coronary artery calcification; eGFR: estimated glomerular filtration rate. All models were adjusted for age and systolic blood pressure. Odds ratio is expressed as a one standard deviation increase in the independent variable (lumbar spine BMD = 0.124 g/cm^2^, lumbar spine T-score = 1.0, right femoral neck BMD = 0.065 g/cm^2^, right femoral neck T-score = 0.9, right total hip BMD = 0.095 g/cm^2^, right total hip T-score = 0.7, left femoral neck BMD = 0.069 g/cm^2^, left femoral neck T-score = 0.6, left total hip BMD = 0.096 g/cm^2^, and left total hip T-score = 0.7).

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
