# Peer review of "Association of Bone Mineral Density and Coronary Artery Calcification in Patients with Osteopenia and Osteoporosis"

_diagnostics, 2020, doi:10.3390/diagnostics10090699_

Round 1

Reviewer 1 Report

The manuscript from Chuang et al investigated the association between bone mineral density and coronary artery calcification in participants with osteopenia or osteoporosis. The study is well designed and the presentation is very straight forward. I have relatively minor question/comments.

  • Did the authors have any detail about the use of pharmaceuticals? For example, were any of the participants taking drugs for osteoporosis or high blood pressure?
  • When were the clinical variables, such as glucose and eGFR, taken with respective to the assessment of BMD and CT?
  • For the diagnosis of osteoporosis and osteopenia, did the authors designate participants as osteoporotic if they had a T-score less than 2.5 at either the hip or spine?
  • Can the authors clarify how the clinical variables were assessed? Certainly these weren't obtained from interviews with the participants?
  • Are the adjusting variables different in tables 3 and 5 and if so, why? Why wasn't glucose, age, BMI, etc included in the models that were limited to participants with osteoporosis?
  • The authors seem to suggest that one of the results for the differences noted in the associations between BMD and CAC in osteoporosis and osteopenia is related to plaque compositional changes with age. If that is true, are the participants with osteoporosis older than those that were osteopenic?

Author Response

Dear reviewers,

        Thank you very much for your invaluable comments regarding our manuscript. We greatly appreciate the time and efforts you spent in reviewing our manuscript. Herein, we submit our revised manuscript and our point-by-point responses to your comments and suggestions.

We hope that our revised manuscript will satisfy your comments and merits publication in Diagnostics.

Sincerely,

Yuh-Feng Wang, M.D., Ph.D.

Reviewer 1, Comment # 1:

The manuscript from Chuang et al investigated the association between bone mineral density and coronary artery calcification in participants with osteopenia or osteoporosis. The study is well designed and the presentation is very straight forward. I have relatively minor question/comments.

Response to Reviewer 1, Comment # 1:

We greatly appreciate Reviewer 1 for all valuable comments and suggestions, which helped us to improve the quality of the article.

------------------------------------------------

Reviewer 1, Comment # 2:

Did the authors have any detail about the use of pharmaceuticals? For example, were any of the participants taking drugs for osteoporosis or high blood pressure?

Response to Reviewer 1, Comment # 2:

Unfortunately, we did not have the information about the participants’ pharmaceuticals. Nevertheless, laboratory data, instead of their associated diseases, were used for the adjustment of potential confounding effects in the multiple regression model. We believe this should minimize the effects of disease treatment on disease status and the physiological condition of the participants. We have mentioned this point in the limitation section (Line 229-232).

------------------------------------------------

Reviewer 1, Comment # 3:

When were the clinical variables, such as glucose and eGFR, taken with respective to the assessment of BMD and CT?

Response to Reviewer 1, Comment # 3:

The clinical variables of the participants were collected on the same day as their assessment of BMD because both measurements were parts of the health examination. We add the information in the section of “2.4. Measurement of demographic and clinical variables”. (Line 76-77).

------------------------------------------------

Reviewer 1, Comment # 4:

For the diagnosis of osteoporosis and osteopenia, did the authors designate participants as osteoporotic if they had a T-score less than 2.5 at either the hip or spine?

Response to Reviewer 1, Comment # 4:

Yes. According to the ISCD/IOF (International Society for Clinical Densitometry/ International Osteoporosis Foundation) for the diagnosis of osteoporosis and osteopenia, participants were designated as osteoporotic if they had a T-score equal or less than 2.5 at either the hip or the spine. We added the information in the section of “2.3. Measurement of Bone Mineral Density” (Line 71-74).

------------------------------------------------

Reviewer 1, Comment # 5:

Can the authors clarify how the clinical variables were assessed? Certainly these weren't obtained from interviews with the participants?

Response to Reviewer 1, Comment # 5:

We appreciate Reviewer 1 for pointing the need for us to clarify this point. We have completely revised the description in the section of “2.4. Measurement of demographic and clinical variables” to clarify this point as the following:

“Information on demographic variables was ascertained by face-to-face interview or questionnaire on the same day of the BMD assessment. The demographic variables included age, sex, body mass index, and smoking history while medical history included hypertension, diabetes mellitus, and hyperlipidemia. In addition, blood samples were also collected on to same day to obtain data on high-density lipoproteins, fasting blood glucose, calcium, alkaline phosphatase, and estimated glomerular filtration rate (eGFR). Systolic blood pressure was also recorded. (Line 76-81).

------------------------------------------------

Reviewer 1, Comment # 6:

Are the adjusting variables different in tables 3 and 5 and if so, why? Why wasn't glucose, age, BMI, etc included in the models that were limited to participants with osteoporosis?

Response to Reviewer 1, Comment # 6:

We thank Reviewer 1 for raising this point. The adjusting variables in Table 3 and 5 were selected based on the results in Table 2 and 4, respectively. Only variables that are significantly associated with CAC were included in Table 3 and 5. One of the requirements for a variable to act as a confounder is that it must be associated with the outcome variable (CAC in our case). In addition, as both hypertension and systolic blood pressure were associated with CAC, we selected the continuous variable (SBP) instead of the binary variable (hypertension), which we think should reduce misclassification error. Furthermore, the clinical data were obtained from laboratory measurements whereas diseases were collected based on self-report from the participants. We also selected fasting blood glucose instead of diabetes mellitus for the same reasons.

------------------------------------------------

Reviewer 1, Comment # 7:

The authors seem to suggest that one of the results for the differences noted in the associations between BMD and CAC in osteoporosis and osteopenia is related to plaque compositional changes with age. If that is true, are the participants with osteoporosis older than those that were osteopenic?

Response to Reviewer 1, Comment #7 :

Yes. The differences noted in the associations between BMD and CAC in osteoporosis and osteopenia were related to plaque compositional changes with age.

We analyzed our data of these two groups according to your comment. The participants with osteoporosis were older than those who were osteopenic. We have added this data and description in the Discussion section. (Line 211-212)

osteopenia

osteoporosis

p

Age, mean (SD), years

56.4 (9.0)

59.5 (9.7)

0.032

Reviewer 2 Report

The study is impressive; however, there are important issues which should be addressed. In the table I, the No CAC group is composed of 59 female, while the CAC groups are formed of 35 and 31 % females, which should be solved. So moderate and CAC could be comparable. Table 2 has a different number of controls, and there is then a statistics type II error of subdividing groups. In conclusion, if the two groups management and CAC are compared, the general odd ratios can be accepted. Diabetes is a significant marker for severe CAC and is almost not discussed. How many of the hypertensive patients have diabetes? Would the risk sum up? Table 4, does not respond to the question but identifies as hypertension as the highest odd ratio.

Hip BMD can also be an important indicator, but again how many hypertensive/diabetic patients have a higher score?

The discussion and conclusions can be improved.  

Author Response

Dear reviewers,

       Thank you very much for your invaluable comments regarding our manuscript. We greatly appreciate the time and efforts you spent in reviewing our manuscript. Herein, we submit our revised manuscript and our point-by-point responses to your comments and suggestions.

We hope that our revised manuscript will satisfy your comments and merits publication in Diagnostics.

Sincerely,

Yuh-Feng Wang, M.D., Ph.D.

Reviewer 2, Comment # 1:

The study is impressive; however, there are important issues which should be addressed. In the table I, the No CAC group is composed of 59 female, while the CAC groups are formed of 35 and 31 % females, which should be solved. So moderate and CAC could be comparable. Table 2 has a different number of controls, and there is then a statistics type II error of subdividing groups. In conclusion, if the two groups management and CAC are compared, the general odd ratios can be accepted.

Response to Reviewer 2, Comment # 1:

We appreciate Reviewer 2 for raising this point. The differences in the sex ratio in the three CAC groups were adjusted for in Table 3. The reason that we did not adjust for sex in Table 5 is that it did not fulfill one of the requirements to act as a confounder (Sex was not significantly associated with CAC).

The number of patients in the three CAC group (no CAC, moderate CAC, and high CAC) in Table 2 was different from those in Table 1 is because only patients with osteopenia was included in Table 2. The number in Table 2 and 3 were the same. For Table 4 and 5, the number represents those with osteoporosis.

------------------------------------------------

Reviewer 2, Comment # 2:

Diabetes is a significant marker for severe CAC and is almost not discussed. How many of the hypertensive patients have diabetes? Would the risk sum up? Table 4, does not respond to the question but identifies as hypertension as the highest odd ratio.

Response to Reviewer 2, Comment # 2:

We agree with Reviewer 2 that diabetes appeared to be significant factor associated with severe CAC. However, diabetes was evaluated as a confounder, rather an independent variable of interest, in the association between bone mineral density and CAC in the present study. Similar, hypertension was considered as a confounder in the association of interest. We adjusted systolic blood pressure instead of hypertension in Table 5 because the former was a continuous variable, which is likely to have a lower risk of misclassification compared with hypertension. In addition, the clinical data were obtained from laboratory measurements whereas diseases were collected based on self-report from the participants.

In table below, we tabulated the data of the participants with hypertension and diabetes mellitus.

Diabetes mellitus (+)

Diabetes mellitus (-)

Total

Hypertension (+)

13 (22.8%)

44 (77.2%)

57 (100%)

Hypertension (-)

181 (4.2%)

181 (95.8%)

189 (100%)

Regarding the comment on whether the risk of diabetes and hypertension sum up in patients with osteoporosis, we are unable to perform any formal analysis because there is only one patients with have both hypertension and diabetes in this group.

------------------------------------------------

Reviewer 2, Comment # 3:

Hip BMD can also be an important indicator, but again how many hypertensive/diabetic patients have a higher score?

Response to Reviewer 2, Comment # 3:

According to our results in Table 3 and 5, hip BMD is unlikely to become significant given the observed p values. The potential confounding effect of hypertension and diabetes were adjusted using systolic blood pressure and fasting blood glucose.

The table below shows the number of hypertensive/diabetic patients with no CAC, moderate CAC, and high CAC.

Variable

No CAC

Moderate CAC

High CAC

p

(score = 0)

(score = 1–100)

(score ≥ 101)

N (%)

142 (57.7)

62 (25.2)

42 (17.1)

Hypertension (%)

18 (12.7)

15 (24.2)

24 (57.1)

<0.001

Diabetes mellitus (%)

6 (4.2)

3 (4.8)

12 (28.6)

<0.001

Hypertension and diabetes mellitus (%)

4 (2.8)

1 (1.6)

8 (19.0)

0.001

Hip BMD could not be compared directly due to different age and sex. Therefore, we evaluated the hip indicator by T-score. In the table below, we analyzed the number of three levels of right/left femoral neck T-score in hypertensive/diabetic patients.

Right neck T-score

Equal and more than -1.0

-1.0 > T-score > -2.5

Equal and less than -2.5

Total

Hypertension

N (%)

12 (21.1)

37 (64.9)

8 (14.0)

57 (100)

Diabetes mellitus

N (%)

5 (23.8)

14 (66.7)

2 (9.5)

21 (100)

Left neck T-score

Equal and more than -1.0

-1.0 > T-score > -2.5

Equal and less than -2.5

Total

Hypertension

N (%)

16 (28.1)

36 (63.2)

5 (8.8)

57 (100)

Diabetes mellitus

N (%)

9 (42.9)

11 (52.4)

1 (4.8)

21 (100)

------------------------------------------------

Reviewer 2, Comment # 4:

The discussion and conclusions can be improved. 

Response to Reviewer 2, Comment #4:

We added a discussion of the pathophysiological mechanism between CAC and BMD, as below: “The pathophysiology is similar between vascular calcification and osteogenesis [3,37]. Vascular calcification has been linked to dysregulated bone and mineral metabolism, reduced BMD and increased fracture rates [5-7]. Vascular calcification is recognized as an active process involving an interaction between inducers and inhibitors [4], and it represents a complex biological process of calcium phosphate deposition and is related to the regulation of osteogene expression, bone morphogenetic protein, calcification inhibitors and inflammatory cytokines [39,40]. (Line 222-227). We have also revised the references accordingly.

In addition, we added in Conclusions section the following sentence: “Although CAC has complex regulatory networks, once osteoporosis develops, CAC needs to be taken care of.” (Line 240-241)

Round 2

Reviewer 2 Report

Thank you for answering the queries. However, I would request to incorporate in the supplementary files the tables presented to respond to my queries. They are important along with the comments. 

Author Response

Reviewer 2, Comment # 1:

Thank you for answering the queries. However, I would request to incorporate in the supplementary files the tables presented to respond to my queries. They are important along with the comments.

Response to Reviewer 1, Comment # 1:

We greatly appreciate Reviewer 2 for re-reviewing our manuscript and providing the valuable suggestions, which helped us to improve the quality of the article.

We have added the following paragraph in the Discussion section: “Furthermore, although diabetes is an important marker for severe CAC, we did not include it in the multiple regression model. Instead, we evaluated fasting blood glucose, systolic blood pressure, and high-density lipoproteins for their potential confounding effects. The latter continuous variables were used because they were obtained from laboratory measurements, whereas diseases were collected based on self-report by the patients. We believe the use of laboratory results would be associated with a lower risk of misclassification.” (Line 195-200). We have also added a new row in Table 1 showing the results of patients with both hypertension and diabetes mellitus. In addition, we have added the following table as Supplementary Materials in the revised manuscript to show the distribution of T-score in patients with hypertension or diabetes mellitus. (Line 102-104)